# Conceptualization of Biomimicry in Engineering Context among Undergraduate and High School Students: An International Interdisciplinary Exploration

**DOI:** 10.3390/biomimetics8010125

**Published:** 2023-03-17

**Authors:** Ibrahim H. Yeter, Valerie Si Qi Tan, Hortense Le Ferrand

**Affiliations:** 1National Institute of Education, Nanyang Technological University, 1 Nanyang Walk, Singapore 637616, Singapore; 2School of Mechanical and Aerospace Engineering, Nanyang Technological University, 50 Nanyang Avenue, Singapore 639798, Singapore; 3School of Materials Science and Engineering, Nanyang Technological University, 50 Nanyang Avenue, Singapore 639798, Singapore

**Keywords:** STEM education, conceptualization, biomimicry, undergraduate and high school students, interdisciplinary

## Abstract

Biomimicry is an interdisciplinary design approach that provides solutions to engineering problems by taking inspiration from nature. Given the established importance of biomimicry for building a sustainable world, there is a need to develop effective curricula on this topic. In this study, a workshop was conducted twice in Singapore: once with 14 students from a local high school in Singapore, and once with 11 undergraduate students in engineering from the United States. The workshop aimed to better understand how students conceptualize biomimicry following the bottom-up and top-down biomimetic methods. The workshop contained a lecture and laboratory session, and data were collected via questionnaires, field observation, and participant presentations at the end of the laboratory session. A qualitative analysis revealed that the top-down biomimetic approach was initially understood using vague and generic terms. In contrast, the students described the bottom-up approach using precise and technical vocabulary. By naming the themes highlighting the students’ conceptualizations, it was concluded that strengthening the principle that makes the natural object unique and increasing interdisciplinary knowledge are needed to help them perform the top-down approach. The results from this work should be confirmed with a more significant number of participants, and they could help develop a curriculum to teach the two approaches effectively by providing tools to help the students generalize their ideas and abstract meaning from systems.

## 1. Introduction

Biomimicry is a process that “borrows ideas from nature” [1] to find creative and sustainable solutions to human challenges [2,3]. Biomimicry has similarities to biomimetics, which is the “interdisciplinary cooperation of biology and technology or other fields of innovation with the goal of solving practical problems through the functional analysis of biological systems, their abstraction into models, and the transfer into and application of these models to the solution”, according to the standard ISO 18458. Therefore, biomimicry is promising for addressing the problems of the 21st century [4]. To date, biomimicry has led to a number of technological advances in a large variety of domains, including industrial design [5], construction [6], advanced materials [7], and biomedicine [8], among others. Many innovations have made it to real commercial applications, such as the bullet train in Japan, Velcro, the Shard tower in London, etc. In addition to the technological outcomes, biomimicry sparks new and sustainable ideas, such as developing scientific research that could be feasible in low-resource labs, providing avenues for new and valuable functionalities, and bridging gaps across technological and cultural knowledge [9]. Contrary to more traditional scientific research, biomimicry tends to focus on environmental and social impacts while still providing performance and economic outputs by, for example, using resources more efficiently or by providing solutions that are adapted to their surrounding contexts [6,10,11]. However, the implementation and use of biomimicry are not straightforward, and it poses the question of how it can be effectively integrated into science curricula to nurture and train the engineers of tomorrow.

Educational institutions have recognized the growing urgency to instill an appreciation for the natural environment in their students [12] and to develop education for sustainable development (ESD) that follows the Sustainable Development Goals (SGDs) set by the United Nations in 2015 [13]. The competencies and skills that the future generations learn should also follow and adapt to a rapidly changing world. These skills, such as learning to learn, multiliteracy, and sustainability, should be integrated into new curricula [14]. At the same time, the learning opportunities should be more equitable across the youth and prepare them to be future leaders of change [15,16]. Moreover, the need for an undergraduate education that trains students to solve problems and communicate, transfer knowledge, and collaborate across the technical and nontechnical fields has been pointed out [17]. Teaching younger generations about biomimicry could address some of these points among the many curriculum changes that can be envisaged. Indeed, biomimicry can help students develop teamwork, scientific knowledge, and environmental awareness [18]. For example, at the European level, the BioS4You project, which started in November 2019, aims to teach bioinspired STEM topics to the next generation to produce skilled individuals who can tackle and provide sustainable solutions to address the grand challenges of the century [19]. However, there is a lack of awareness and professional knowledge, training, and expertise regarding biomimicry that dramatically hinders the development of sustainable technological solutions and education worldwide [20,21].

To develop relevant curricula on biomimicry and teach them effectively, it is necessary to consider how students learn, comprehend, and process the subject matter. Although literature and report reviews on universities and institutions that teach and train biomimicry to exist, mainly U.S. and European higher education institutes have been studied [22]. Furthermore, biomimicry is exciting because the principles can be used by people of all ages, education levels, backgrounds, and cultures (including minorities) [23,24]. However, a cross-age and cross-cultural study on how they are conceptualized has yet to be undertaken. Such a conceptualization can guide the development of a new curriculum by highlighting gaps, misconceptions, and other learning responses to particular teaching material. Conceptualization studies are widely used in educational sciences, especially in the areas of science and technology [25], the construction of evidence [26], and academic self-concept [27].

This study explores how students conceptualize biomimicry, focusing on what they categorize as the problem, the natural system chosen, and how to apply that naturally existing solution to an engineering problem. First, observations from several workshops and courses to examine students’ learning and their conceptualization of biomimicry are summarized in a literature review. This review revealed two significant approaches to the biomimetic process: the top-down and bottom-up approaches. Then, a workshop was conducted with two groups of students from different cultural backgrounds, ages, and education levels. The workshop setup, the methodology for the analysis, and the objectives are described. The data were then analyzed: first, the students’ predisposition to the workshop, then their conceptualization of the top-down and bottom-up approaches of biomimetics, and finally, their appreciation of the workshop. A discussion and implications, conclusions, and future work follow the analysis of the results. Overall, the following research questions are addressed:-How do the participants conceptualize biomimicry in the case of the top-down approach?-How do the participants conceptualize biomimicry in the case of the bottom-up approach?

## 2. Literature Review

### 2.1. Methods and Tools to Apply Biomimicry

There are two main approaches to applying biomimicry: the bottom-up method, also called the solution-driven method or biomimicry, and the top-down method, also called the problem-driven method or indirect biomimicry [28,29,30]. In the bottom-up approach, the natural biological organism directly inspires the technological innovation following three steps: (i) qualitative and quantitative analyses of the biomechanics and functional morphology of a biological system; (ii) the abstraction phase, in which ideas generated from the biological model are separated; (iii) the implementation of the solution to the technical system. In the top-down approach, biomimetic improvements and innovations are applied to pre-existing technical products also following three steps: (i) the identification of the technical challenges and boundary conditions; (ii) the search for natural examples of solutions that might serve as models for resolving the technical issue; (iii) the differentiation of the obtained solutions from their natural examples through abstraction before their technical implementation. Speck et al. also derived an extended top-down process in which biomimetic research and development follow the same steps as regular top-down projects, including multiple iterations to incorporate the most effective biological templates into the technical product [29].

Biomimicry is an interdisciplinary field in which biology and engineering interact and discuss at various levels of abstraction and within several iteration cycles. Several frameworks and tools have been developed to help engineers find optimum biological designs. Such frameworks are based on structure–behavior–function (SBF), which is a programming language for engineers [31], and the concept-knowledge (C-K) theory, which integrates multiple domains of information to facilitate the making of connections and innovations [32], and the exploration–design–implementation method [33]. In terms of tools, the most frequently used tools are biological design repositories such as AskNature.org or BioTRIZ [34], Ashby diagrams [2], general finite element method software such as ANSYS, Abaqus, and COMSOL Multiphysics [35,36,37], and keyword search tools for students or engineers to search for the appropriate biological model [38,39,40].

Several workshops and courses have been implemented in curricula at various education levels based on these methodologies, frameworks, and tools. The field observation of the students’ learning is reviewed in the following section.

### 2.2. Students’ Learning

In the context of education, we found several definitions of biomimicry that do not necessarily follow the ISO standard. To make this study comprehensive of the field, we had to broaden our view of the definition of biomimicry and consider other terms, such as “bioinspiration.” According to ISO standard 18458, bioinspiration is “a creative approach based on the observation of biological systems, the relation to the biological system may only be loose.” Other related terms, such as “biologically inspired” or “inspired by nature,” were also found and reviewed. Based on this observation, the following literature review was conducted using three keywords to find published papers on students’ learning experiences after taking a module relative to biomimicry. One keyword used was related to the subject of the module: “biomimicry” or “bioinspiration.” Other keywords were related to the learning: “curriculum,” “education,” “course,” or “workshop,” and another group of keywords were related to geographical regions or countries: “Europe,” “Asia,” “Italy,” etc. The literature review is not exhaustive, as teachers and lecturers do not always publish their course development and experiences. Nevertheless, the number of papers gathered (i.e., 28) provides a representative overview of the current knowledge on the teaching of biomimicry or related terms (Figure 1a). The increased published research output in the past decade illustrates the growing interest in this topic (Figure 1b).

Existing studies have employed biomimicry concepts through coursework and/or workshops with a diverse range of topics; for instance, innovative material engineering based on biological diversity [41], nature as a model, measure, and mentor [42], biologically inspired design [43], integrating biology and design for sustainable innovation [44], and a biomimicry robot education program [45]. Most curricula had a lecture component, in which the principle of biomimetics was explained with examples, and an active component, in which the students or pupils applied the principles in context. A few curricula were only a series of seminars, and a few others were project-based. The modules were either as long as a semester or as short as a couple of hours. The participants’ learning experiences were evaluated using questionnaires, surveys, field notes, and interviews.

The published studies of these modules also show diversity in terms of the participants’ culture, age, and education (Figure 2). Most of the studies were conducted in North America and Europe, with the United States having the most published studies in this area (Figure 2a). Additionally, other studies have been conducted in Europe (i.e., Germany, the Netherlands, and Turkey), Africa and the Middle East (i.e., South Africa and Israel), and Asia (i.e., Japan, Korea, and Malaysia). The participants of these studies were predominantly undergraduate university students, although all age groups can be found (Figure 2b). It is interesting to note that Asian and Middle East countries tended to study young children and people of all ages. In contrast, Northern America and Europe mainly focused on university students. People of different age groups were taught about biomimicry through exhibitions in museums or at zoological institutions [41,46,47]. For university students most of the modules were conducted for engineering students, including mechanical engineering, industrial design engineering, and biomedical engineering (Figure 2c). Only one study reported teaching a cohort of students from multiple disciplines: mechanical engineering, industrial and systems engineering, material science engineering, biomedical engineering, and biology [43]. The other fields outside engineering were liberal arts and science, graphic design, and architecture.

Despite the diversity of the students’ and module participants’ backgrounds, ages, and cultures, the outcomes of the studies did not show any significant differences. Most papers reported similar positive outcomes, and the remaining challenges are reported in Table 1. The significant challenge that has been reported many times is the difficulty in communication and the analogies between the technological and biological fields [48], which call for the development of keywords and other search tools to facilitate teaching [30,49].

This brief literature review suggests that improvements are still being made to facilitate students’ learning and application of biomimicry. In particular, one remaining gap is related to better understanding the conceptualization and ideation process of the students during both the top-down and bottom-up approaches of biomimetics. The following sections of the current study are concerned with addressing this gap.

## 3. Methodology

### 3.1. Workshop Structure

The participants attended a four-hour workshop on biomimicry titled “Taking inspiration from Nature to engineer tomorrow’s world.” The workshop consisted of two main parts: a lecture (Part 1) and an active session (Part 2), which were used to test the student’s responses to the top-down (Part 1) and bottom-up (Part 2) approaches. The lecturer conducted the lecture while the participants were seated in groups of 3 to 5 around round tables. The active session consisted of laboratory experiments and a PowerPoint (PPT) slide preparation and sharing them with the class in a 5 min presentation. The laboratory session was facilitated by postgraduate students and researchers working on these topics. The details of the two workshop parts are given in the following sections. The PPT slides of the workshop are provided in the Appendix A. The workshop was preceded and concluded by surveys to induce reflection and self-assessment in the participants.

#### 3.1.1. Preworkshop

At the start of the workshop, a few minutes were taken for the participants to complete a survey. In this survey, the participants had to rate, on a scale from 0 to 10, their degree of confidence (i.e., belief in their current ability), their motivation, how successful they would be, and their degree of anxiety (i.e., how apprehensive they would be) about performing the following tasks: create or engineer a design; identify a design need; research a design need; develop design solutions; select the best possible designs; construct a prototype; evaluate and test a design; communicate a design; redesign; consider the environment and sustainability in the design; apply engineering tools; work as a team; manage the project (deliverables, deadlines, etc.)

#### 3.1.2. Workshop Part 1: Lecture

During the lecture in the first part of the workshop, the participants were given a brief introduction to biomimicry. The lecturer asked the students Question 1: Can you give some examples of biomimicry around us and your thoughts about how this can be useful? The lecturer then taught the students the top-down approach, or problem-based approach, of biomimetics using the following three steps:Determining the problem to solve/aims (e.g., Can we use the same material on Earth, which is a hot environment, and in space, which is a cold environment?);Identifying a natural system in which the problem is solved and understanding the concepts (e.g., Is there any natural species that live in a hot and cold environment and how does it do it?);Developing a method to apply this solution to the engineering problem (e.g., If the identified species could live in a range of temperature from −30 °C to +10 °C, can we adapt the strategy and mechanism identified to engineer a solution that can work from −100 °C to +20 °C?).

The example that illustrates this approach was purposely complex to suggest the high level of abstraction in the biomimetic process, as there is no “nature” in space. The lecturer then provided three concrete examples of applying the methodology.

The first example was the Esplanade, a Singaporean building that mimics the shape and structure of the durian fruit [60]. After showing this example, the lecturer detailed the three steps of the top-down approach in designing the Esplanade. Then, the students were introduced to two additional scenarios via videos (A and B), and after that, questions were asked. Video A explained how Japan’s Shinkansen bullet train was redesigned based on the aerodynamic features of three birds (i.e., the kingfisher, owl, and penguin) to address the problems of noise pollution and train efficiency [61]. Video B showed Aquaporin, a Danish water technology company with an office in Singapore that developed water-filtering membranes using aquaporin proteins, which are water channels at the surfaces of all living things [62]. After each video, the students explained how biomimicry was employed by answering the following three questions based on the top-down biomimetic process:Question 2a: What is the problem addressed?Question 2b: What natural system was chosen?Question 2c: How did they apply the natural solution to the engineering system?

#### 3.1.3. Workshop Part 2: Laboratory Session

After the lecture, the students were gathered in five groups of two to three; each group had a biological sample as a theme. The biological samples were edamame, lotus leaf, seashell, peach gum, and mushroom. The facilitators, who were researchers, working on engineering materials that shared features with these biological samples, introduced their respective groups to these biological elements and explained why they were studying them. Then, the facilitators conducted their groups in the laboratories for the students to conduct hands-on experiments. Although each group had a different biological sample and performed different experiments, the idea for the laboratory experience was to understand how the traits of the biological samples could be used to develop advanced materials and technologies. This part of the workshop addressed the other approach of biomimetics: the bottom-up approach. A summary of the laboratory experiments is provided in Table 2.

Before going to the laboratory, the students were instructed to take notes and pictures to complete a PPT slide template. After completing their laboratory experiments, each group presented their slides to their class to learn from each other. In the template, the students were instructed to explain their initial thoughts when they observed the natural specimen provided (Question 3a: What initial thought came to mind when seeing the object?). Then, they had to explain what they did during the laboratory experiment and what they learned (Question 3b: What did you do in the lab, and what key concept did you learn?). Finally, they had to explain for what application they could use the mechanism from the biological specimen and which problem it could solve (Question 3c: For what application could the mechanism be learned to be used?) The presentation typically lasted three to five minutes and was followed by questions and answers.

#### 3.1.4. Postworkshop

After the workshop, the participants were requested to provide comments on the workshop and to give some appreciation of it.

### 3.2. Participants

Two groups of students participated in the workshops in May 2022 in Singapore. The workshop was conducted twice: once for 11 undergraduate (UG) students from the School of Mechanical Engineering at Purdue University in the United States and once for 14 preuniversity high school (HS) students from various Singaporean junior colleges. The materials provided to both groups of participants during the workshops were identical. In each workshop, the same two scenarios were provided (Scenario A: Japanese Shinkansen bullet train and Scenario B: aquaporin water filtration), and the same laboratory experiments were conducted. The undergraduate students’ ages ranged from 20 to 23 years, and the high school students’ ages ranged from 16 to 18 years. The gender balance was 38.9% for undergraduate students and 66.7% for high school students. In the workshop context, the pool of participants comprised students who were motivated and interested in learning. Indeed, the Purdue students were engaged in a study-abroad program about biomimicry, which led them to a trip to Singapore for a few weeks. Similarly, the local high school students were invited to discover the university through the module of their choice. Therefore, the students chose to join the workshop of their own accord.

### 3.3. Data Collection

Data were collected through online surveys using a Google form and Wooclap, an interactive digital platform that creates questionnaires. The field notes, pictures, and slides (or PPT slides) that the students prepared during the workshop were also collected. The data gathered were aimed at answering the 4 research questions summarized in Table 3.

To collect data and answer the research questions, two sets of responses were collected: one from undergraduate students and one from high school students. The participants responded to the questions on their smartphones or laptops after receiving a link or QR code. After the workshops, the participants’ survey responses were exported to a spreadsheet for data analysis. The pictures, notes, and final PPT slides prepared by the students were collected during and after the workshop.

### 3.4. Data Analysis

Several analyses were conducted, both quantitative and qualitative. The quantitative analysis of the preworkshop survey was conducted based on the data exported in MS Excel. The average grades for each question were obtained for the two groups of participants. Qualitative inductive analysis was conducted for the Part 1 questions of the workshop, including the answers to Question 1 and the series of Questions 2a, 2b, and 2c. For Question 1, the responses were compared to the words in the word cloud published in ref [28]. For Questions 2a, 2b, and 2c, independent coders coded the qualitative data responses from which the themes were identified. Each unique code was categorized under a broader theme. These themes identified the specific topics, ideas, and patterns that repeatedly emerged from the data. It should be noted that the sum of labels reflected in Table 1 and Table 2 may differ from the actual number of participant responses, as a single response from a participant may have been assigned two different labels (e.g., both “noise” and “efficiency” were counted separately). Additionally, some participants failed to respond adequately to the question. Part 2 of the workshop and the postworkshop feedback results were analyzed qualitatively. The data presented are labeled with the students’ numbers for the survey responses, or the students’ groups (roman numbers from i to v) for the laboratory experiment, and with their educational background: HS for high school and UG for the undergraduate student.

## 4. Results

### 4.1. Predisposition of Students to Combining Design and Sustainability

Before conducting the workshop activities and analysis, it was essential to obtain some context on the students’ interest in and predisposition to learning and understanding the concepts of biomimetics. The preworkshop survey and Question 1 were conducted to this aim. The data were analyzed and are reported in Figure 3 in the form of radar plots.

Although two populations of students were tested, this work did not aim to compare the students but to obtain a global understanding of how they conceptualized biomimicry. The preworkshop survey revealed some differences between the two groups, with the Western undergraduate students showing higher degrees of confidence, motivation, and successfulness expectation than the Singaporean high school students for a lower level of anxiety for all the topics surveyed. Yet, both groups showed moderate confidence, motivation, and success (above 6 out of 10 and below 8). This finding indicates that, on average, the participants had positive perspectives and attitudes concerning design, engineering, teamwork, etc. However, the degree of anxiety ranged, on average, below 5 for undergraduate students and above 5 for high school students, suggesting that education and training might increase confidence and motivation while reducing anxiety in engineering students. Indeed, the undergraduate students were in their third and fourth years of university studies, whereas the high school students were still unfamiliar with the university curriculum. Several educational studies have reported a significant increase in confidence from Year 1 to Year 4 in undergraduate students, which would explain what was observed here [63,64]. This is particularly visible in the degree of confidence and degree of anxiety, where the differences in the average scores between the two groups are two or greater. In contrast, their motivation and expectation of success differ in scores of only one, on average.

The participants provided some interesting ideas when asked about their views concerning biomimicry (Table 4). These ideas were similar to what has been reported in other studies on biomimicry and in word clouds [28]. Interestingly, the proposed ideas could be ordered into two significant categories: nature-oriented and technology-oriented applications (Table 4). This classification closely resembles the two methods of biomimetics, where nature-oriented propositions focus on the biological organism to obtain the idea, such as in the bottom-up process, and technology-oriented propositions focus on the technological implications, such as in the top-down process. For each of the ideas reported in Table 4, the biological trait and field of application associated with the idea could be proposed. It is interesting then to note that the nature-oriented ideas have biological inspiration that is identifiable, particularly in that they tend to mimic the shape or mechanism of the biological organism, whereas their field of application is not always apparent, and only a few could be easily attributed.

In contrast, the technology-oriented propositions had less clear bioinspiration traits, and only those identified were relative to increasing the performance, sustainability, or providing new capabilities to the engineered system. However, the field of application was identifiable for most of them. This suggests that although the participants had no prior knowledge about biomimetic methods, they intuitively gained a sense of the processes. However, it also shows a need to clarify how these methods proceed to enable the conscious development of biomimetic solutions.

The preworkshop survey and answers to Question 1 informed us that the participants had a high degree of motivation and an interest in engineering, design, and incorporating sustainability into their design solutions. The results also showed that they had some intuitions and preconceptions about biomimicry and how they could be applied. There was no noticeable difference in the responses between the two groups, despite the differences in the levels of education and cultural backgrounds. Therefore, for the rest of the study, the two groups of participants had the same predispositions to learning biomimicry.

### 4.2. Conceptualization of Top-Down Biomimetic Method

To better understand how students conceptualize the top-down method of biomimetics, we analyzed the answers to Questions 2a, b, and c for Scenarios A and B. Six themes were identified according to the students’ responses (Table 5).

For Question 2a, “What is the problem addressed?”, the two themes that emerged from the answers received are “application” and “performance.” The responses under the theme “Application” identified the problem as the technology’s application, aim, or usage. These responses also seemed to focus more on the ultimate issue faced by the consumer that should be eliminated (e.g., loud noise) or improved or changed in a general sense (e.g., water treatment and transport). Other responses identified the performance of the existing technology as the problem. The performance here was categorized as technology that not only achieves the task, but also performs it better with the help of biomimetics. The answers, therefore, pertain to how good the technology is for the indented application. Here, it seems that the students focused more on the cause of the problem than its usage, which is related to the efficiency of the technology for both scenarios. It is interesting to note that the students seemed more readily focused on and were able to identify the application more often as the problem, which suggests that students are more inclined to consider the problem as its effect rather than its root cause.

To Question 2b, “What natural system was chosen?”, the students’ answers could be categorized into two themes: “general” and “specific.” All the students watched the same videos before answering the questions, yet the students’ responses varied in exactitude and specificity. Some responses referenced natural systems in the general sense (e.g., birds and plants), whereas others referenced specific characteristics, components, or processes of natural systems. To illustrate, in Scenario A, responses under the theme “general” were defined as answers that were non-species-specific (e.g., a beak). In contrast, responses under the theme “specific” were species-specific (e.g., a kingfisher beak). From the two themes that emerged, it can be inferred that the students placed differing importance on the specificity of the natural system chosen. Some students were content with referencing the natural system without making specific distinctions. In contrast, other students believed defining the specific component or species of the natural system chosen to inspire the product was important.

The students responded to Question 2c, “How did they apply the natural solution to the engineering system” in two main ways, which resulted in the themes “mimicking nature” and “product modification.” Relative to the theme “mimicking nature,” the students cited mimicking the form or shape of a natural system (e.g., shaping the nose of a train like the beak of the kingfisher, or mimicking a plant membrane). The responses seemed to show a rather superficial understanding of the application. The students’ understanding seemed limited to the apparent process of biomimetics without referring to specific applications. Relative to the theme “product modification,” the student responses were concerned with simply modifying an existing product as the application of the natural solution (e.g., altering the shape of the head or enhancing the membrane). These responses reflected technical actions: the students might think about applying and implementing knowledge into the engineering system. Responses coded under these themes seemed to think one step further than the responses under the “mimicking nature” theme, which was confined to the abstract step. The critical difference between these two themes might lie in the students’ interpretation of the word application. Students either focused more on how the natural system was applied on a more conceptual level, which was reflected in the theme “mimicking nature,” or on how the natural system was applied on a more technical level to the end product, which was reflected in the theme “product modification”.

Analyzing the students’ responses to the three questions on the top-down biomimetic approach revealed six themes under which the answers could be classified. These themes help in understanding how the students conceptualized the top-down biomimetic process. In this process, a technical problem is first pointed out. A biological organism in which the solution is solved is identified before this solution is transposed and applied to solve the technical problem. The six themes, therefore, highlight the difficulties for students in delineating the exact technical problem to solve and describing it in qualitative or quantitative terms, the challenge of identifying what will serve as the biological model (the biological mechanism, shape, design, etc.), and the challenge of transposing the solution to solve the technical problem.

### 4.3. Part 2: Laboratory Session

During the laboratory session, the participants experienced the bottom-up approach of biomimetics. To obtain insight into how the students conceptualized the process, their attitudes during the laboratory session and the PPT slides they generated and shared with the whole group at the end of the workshop were analyzed.

First, it was noted that all the students were engaged during the laboratory session. Working in small groups allowed easy exchanges between the participants and facilitators. Their attitudes demonstrated their engagement: performing the experiments, asking questions related to the experiment they were offered to perform, as well as general questions related to the machines and other pieces of equipment present in the laboratory, and to life at the university. The students’ engagement was also evident during the presentation, during which decorative and funny images were added to the slides, and pictures and videos taken during the laboratory experiments were also shown. Several students presented confidently even as a little play (“What if you take this seashell and try to smash it?”).

Second, analyzing the students’ answers to Questions 3a, 3b, and 3c provided insights into their conceptualization of the bottom-up process. Six themes were identified according to the students’ responses (Table 6).

For Question 3a, the answers could be classified under two main themes: descriptive answers and comparisons with existing objects or places. The descriptive answers generally tried to describe the exterior aspects of the natural samples given to the students (see Table 2 for the list of biological samples), as well as the properties of the sample pieces. These answers suggest an inquiry thought process during which the observer collects clues and details about the object. The comparisons with existing objects connected the biological samples with other existing samples, which aided in finding suitable applications. Some of the answers also highlight the lack of technical vocabulary to describe what is seen efficiently, which explains the use of comparisons or “naïve" description vocabulary.

After performing the experimental tests guided by the facilitators in the laboratory, the students were asked to answer Question 3b about what they learned in the lab. All the answers were very technical and reported either the technical/scientific equipment or methods or technical/scientific concepts. Moreover, the words “properties” and “materials” were reported more than ten times often several times within one slide. This indicates that the students already understood that the applications would be largely based on the properties of the materials. The concepts explained to the students, such as hydrophobicity, self-shaping, anisotropy, and using a living organism (a fungus) to create a material, were often explained to the class in the form of definitions. These observations demonstrate the knowledge acquisition of each group of students and the shift from a general to a specialized language. The tools and methods were also extensively described and pictured as enablers of the technology.

Finally, Question 3c applied what was learned during the experiment to address a real problem or application. Two central themes emerged from the answers: a concrete and general application was given, or an actual means or process was described. Providing the detailed recipe to apply the concept and knowledge they learned suggested the following: (i) an increase in the students’ confidence to apply biomimicry, and (ii) their ability to conceptually transpose what they had seen in the lab to address a more challenging goal. Indeed, the focus for the applications was also more oriented toward transposing the concept, such as “create the same microstructure” instead of copying the shape or other superficial features of the natural samples. It is also noticeable that the natural elements were not mentioned anymore at this stage. Finally, many students also referred to these applications as “potential applications,” with some also mentioning some limitations that would need to be overcome. For example, one group mentioned a means of implementation: “Heating it up and molding the material around the place in need of sealing. The caveat will be that the surface cannot be wet upon application of the gum.” Thinking about the limitations that might lie ahead also demonstrates the thought process of the students, who already envisioned the material in the application and the further issue to solve.

The analysis of the students’ answers to the questions suggests the critical role played by the time in the laboratory learning about the technical concepts and the tools and methodologies that can be used. Within a short time, all the groups of students demonstrated a shift in their vocabularies from general to more specific and showed the ability to make projections toward real applications. Maybe due to the somewhat familiar environment (the laboratory) for the students, or maybe because they were being provided with tools to use, the students seemed to be able to imagine concretely how to apply their conceptual knowledge outside the frame of the workshop, and how to apply biomimicry in a variety of contexts.

### 4.4. Postworkshop

Finally, the postworkshop survey informed the students’ experiences of the workshop. Although the participants’ responses were lesser in number, more than 90% of the answers received indicated that they enjoyed the workshop, while 10% were neutral. All the participants indicated having enjoyed the practical activity and direct interactions with the researcher the most. One participant suggested that “The lab [...] exceeded my expectation with how in-depth the graduate student went into their research and how much care they put into this mini lab experiment” and that “[they] got to do a lot more than [they] expected during the lab”. Some of the challenges reported were regarding the final PPT presentation and coming up with ideas for applications for the mycelium composites. One student pointed out difficulties in understanding some of the concepts. However, the specific concepts they were referring to were not specified. It is interesting to note that these challenges are more related to skills outside biomimicry, such as communication and scientific skills, which further highlights the challenges in teaching and applying the biomimetic processes, as it needs a lot of other interdisciplinary knowledge and skills.

## 5. Discussion

This research is part of a project that aims to develop a curriculum on biomimicry. To do so, we needed to explore how different students conceptualize it. The study looked at how students categorized, applied, and conceptualized the biomimetic knowledge given to them through four-hour in-person workshop participation. The pre- and postworkshop survey analyses, observational data from laboratory sessions, and content analysis on the students’ PPT slides revealed recurring themes that describe their conceptualization of different biomimetic approaches. This section compares our results with the existing literature and comments on the limitations and future research.

### 5.1. Comparison of Bottom-Up and Top-Down Approaches

Analyzing the students’ answers for the top-down (Part 1 of the workshop) and bottom-up (Part 2) approaches revealed different results. Generally speaking, the bottom-up approach of biomimetics seems more readily understood by students who can use traditional engineering tools and methods to abstract and apply biological knowledge to find an engineering solution. While the top-down process was perceived as something vaguer and more general, as pointed out under the themes “General” and “Mimicking nature,” where the students were not able to name or describe the details of a specific organism or how to implement the solution, the bottom-up process was perceived with detail and using specific vocabulary. The general application was instead sometimes absent from the discussion of the students.

These observations have been reported in other studies and may be linked to the lack of knowledge in the field of biology (see Table 1). In the workshop, the two approaches were also taught very differently, with the top-down approach presented via videos and exercises and the bottom-up approach presented during laboratory experiments. Therefore, the active component during the bottom-up approach also engaged the students more, motivated them, and improved their learning. The two different teaching approaches stemmed from the difficulty of developing an active component for teaching the top-down approach, as the tools are essentially databases. A more engaging way to teach this could be to present the students with an engineering problem and elaborate the keywords from this problem to help find a biological species or use one of the available databases. Moreover, it is noticeable that, for the students, the bottom-up approach was easier to grasp because it departed from the natural example. Once the scientific principle that makes the biological sample interesting is stated and defined, such as “anisotropy,” “crosslinking,” or “microstructure,” conventional engineering tools can be used. This suggests that, in future workshops, the “principle” stage should also be reinforced in the top-down approach. The principle is indeed the bridge between the general and the specific.

### 5.2. Need for Interdisciplinarity

Similar to a handful of bioinspiration studies that call for interdisciplinary collaboration [65,66,67], this study also acknowledges the interdisciplinary nature of biomimicry. However, the interdisciplinarity present during our workshop is mainly between the engineering subjects, which are chemistry, material science, and mechanical engineering. Biologists, architects, artists, and social scientists should come on board to teach biomimicry better. Interdisciplinary approaches have proven more effective at motivating students to learn science concepts [66]. In the case of biomimicry, interdisciplinary collaboration can help to strengthen the cross-pollination of disciplines and specific tools, processes, and methods [65], allowing students to acquire more transferable interdisciplinary skills while learning through bioinspiration. However, the integration of multidisciplinary skills still needs to be improved. Two courses were designed in a study developing bioinspired approaches in undergraduate architecture curricula [65]. Both courses took a top-down approach due to a need for interdisciplinary partners that could provide resources for open-ended biological research. The absence of biologists and relevant resources eliminates learning opportunities and limits students’ learning in conducting holistic biomimetic abstraction processes. Future studies should address the need for interdisciplinary skills, which are crucial for allowing students to develop a flexible mindset when adapting biomimicry to any form of problem-solving.

### 5.3. The Need for More Hands-On Learning

As pointed out above, our teaching of both the bottom-up and top-down approaches was asymmetric in the methods due to the lack of time and workforce to facilitate and supervise a more comprehensive workshop. A more active component for teaching the top-down approach is needed. Indeed, in many studies involving student participants, observations and interviews have highlighted that students learn better when there are opportunities to conduct hands-on research and project development activities. For instance, studies have revealed that the combination of presentation, drawing, and reflection helps students to gain more comprehensive learning [60]. In addition, many students said that they needed more time to internalize and carry out the assignments using biomimicry design processes [68]. Students also wished that more aid could be given to help develop their basic ideas into actual prototypes.

Similarly, survey results have shown that when asked what methods could encourage creativity and the practical application of biomimetic design, the most widely voted method by students is “reference buildings, associations, and analogies” [69]. Making direct comparisons with the concept by building a model or making contact with nature aids and enhances creativity in both environmental and ecological aspects. The indication of students’ desire to participate and learn biomimicry through workshops has also been shown through the overwhelming responses in course sign-ups [39]. These observations also confirm our own workshop experience.

### 5.4. Limitations of Study and Future Research

The current study validated some of the observations reported in the literature, providing an added component by comparing the two approaches of biomimetics. It should be noted, however, that the findings of this work are limited to the small sample size of students involved in the study. Additionally, due to the considerable differences between the two groups, such as age, nationality, and level of prior knowledge, the findings of this study are not meant to present a comparison between the two groups. The background and knowledge of the researchers teaching during the workshop may also have influenced the students’ learning. Another limitation of our comparative analysis of the top-down and bottom-up approaches is that the introduction of one before the other puts students in a better position to comprehend the approach that is introduced second. Hence, to achieve a less biased comparative analysis of these approaches, a future study could look at their differing impacts using two separate groups of participants. Finally, another limitation of the study is the lack of reflection on the impact of the students’ prior exposure to the concepts introduced on the effectiveness of the two approaches.

In future work, it would be very interesting to investigate the effect of culture and educational training on the conceptualization and understanding of the students. One common hypothesis is that more education facilitates learning, as more interdisciplinary skills are acquired. However, more education may not necessarily favor abstraction and transposition from a biological sample to an engineering application, as a lot of plasticity and imagination may be required. Furthermore, it would also be interesting to test the hypothesis of whether living closer to nature or to a place where bioinspiration is more readily applied and advertised would help students’ learning. The laboratory experiments could also be better studied and analyzed to understand whether they helped or influenced the degree of abstraction that the students attained. Finally, a remaining question would be how teaching biomimicry influences students’ motivation to pursue STEM careers.

## 6. Implications, Conclusions, and Recommendations

As bioinspiration/biomimicry becomes increasingly recognized as an essential concept and process in STEM, particularly in engineering disciplines, it becomes crucial to explore how students approach the concept and, in particular, its two main approaches. Through thematic analysis, this study systematically reviewed how students approach biomimicry, revealing six themes for the top-down process and six themes for the bottom-up process. These themes revealed the different modes of understanding and ideation of the students depending on the process. We found that the ages and cultures of the students had little impact on their conceptualization of biomimicry. Furthermore, our work highlights the effectiveness of a hands-on approach in increasing both learner motivation and learner outcomes. The impact of varying interdisciplinary knowledge and skills on student understanding of the concepts is also revealed in comparing both the top-down and bottom-up processes of biomimetics. There was a bias in the differing teaching tools used for both approaches that impacted our comparative analysis of their effectiveness. A future study could be conducted using more similar tools for both approaches. There may also be concealed biases due to the lack of understanding of the students’ prior knowledge and its impacts on the results. Nevertheless, because the top-down approach is desirable for engineers, students learning biomimicry in engineering programs would benefit from a top-down approach that is reinforced by a bottom-up approach, and that employs hands-on teaching tools, as our study has shown that the hands-on active approach is more effective at improving students’ conceptualizations. To date, and the best of the knowledge of the authors of this study, this study is the first of its kind to have been conducted. While the small sample size limits the findings of this work, the results offer novel, interesting insights that educators can use when designing biomimicry teaching materials and when teaching high school and undergraduate students biomimicry. This study should be repeated with a larger sample size to obtain more accurate and representative results. Further research should be conducted to fully explore the research questions in this study. For example, future studies could examine the effects of the top-down approach followed by the bottom-up approach, and vice versa, on two different groups of participants to determine whether differing the chronology of the methods impacts participant conceptualizations. In addition, the study might be more effective if it was conducted over more sessions, as the time constraints might impact the students’ ability to fully comprehend the complex subject matter.

## Figures and Tables

**Figure 1 biomimetics-08-00125-f001:**
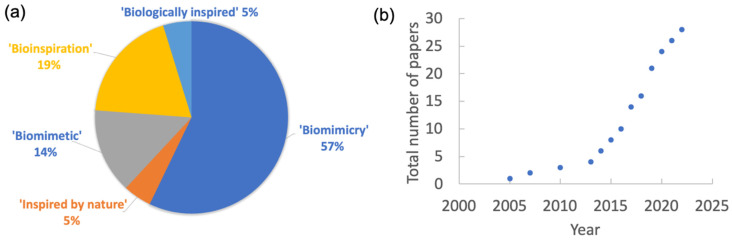
(**a**) Keywords used in published papers that study students’ attitudes and ideation related to biomimicry and their respective percentages. (**b**) The number of published studies is a function of the year of publication.

**Figure 2 biomimetics-08-00125-f002:**
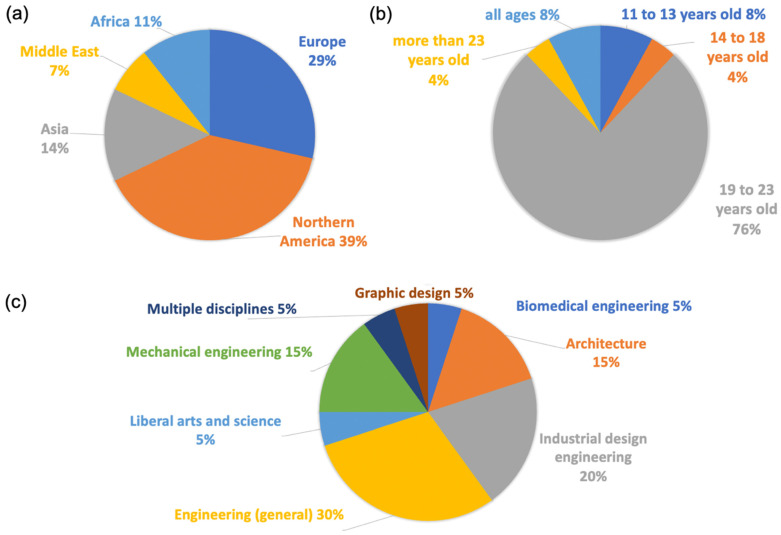
Percentages of published studies classified by (**a**) region, (**b**) age group, and (**c**) major.

**Figure 3 biomimetics-08-00125-f003:**
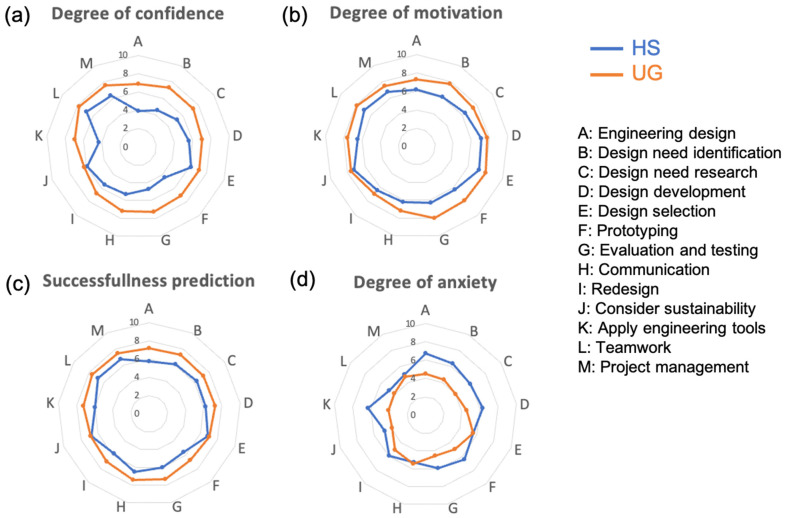
Radar plots showing grades on a scale from 0 (lowest grade) to 10 (highest grade) relative to participants’ (**a**) degree of confidence, (**b**) degree of motivation, (**c**) rate of successfulness, and (**d**) degree of anxiety to achieve the tasks listed in the plot. The grades obtained from the high school students are in blue, and those of the undergraduate students are in orange.

**Table 1 biomimetics-08-00125-t001:** Overview of positive outcomes and remaining challenges reported in the literature on teaching biomimetics.

Positive Outcomes	Remaining Challenges
- Increased literacy and creative, high-order, and design thinking skills [33,39,47,50,51,52]. - Stimulated interest, curiosity, enthusiasm, fascination, and motivation [23,33,39,45,50,53]. - Made the science learning process more active and enjoyable [45,50,54]. - Enabled participants to use pre-existing knowledge and personal experiences [32,54].	- Students found it difficult to “biologize,” to use analogies and mapping to transfer the knowledge from biology to technology or design (lack of interdisciplinary skills) [23,34,43,53,54,55]. - Confusion in the biomimicry or bioinspiration process [51]. - Lexical ambiguity leads to the students needing help with expressing their ideas [56]. - Challenges in working as a team or group [50,54]. - Difficulties in identifying the constraints and criteria, applying the methodology, and implementing the designs [54]. - Perceived as complex and challenging [41,45,57]. - A tendency to make simple, superficial associations [2,55,58,59]. - Difficulty in ensuring equal engagement by all participants [43,44].

**Table 2 biomimetics-08-00125-t002:** Details of active learning session and its learning objectives.

ID	Biological Sample	Laboratory Experiment	Intended Learning Objective
i	Edamame	- Dry and hydrate the natural edamame seedpod in the oven and observe the change in shape. - Prepare an ink containing a polymer, carbon black, and glass fibers for 4D printing using an extrusion-based 3D printer.	- Understand that materials can change shape depending on their internal microstructure. - Learn that 3D printing technologies can be used to create a shape and control the microstructure in synthetic materials.
ii	Lotus leaf	- Measure the hydrophobicity of various surfaces by depositing a drop of water and using a portable optical microscope to measure the contact angle.	- Understand the concept of superhydrophobicity. - Learn how to measure the contact angle and use an optical microscope.
iii	Seashell	- Measure the hardness of seashells and ceramics using a Vickers indenter and an optical microscope.	- Understand the concept of anisotropy. - Learn how to test the hardness of materials.
iv	Peach gum	- Prepare a hydrogel using a chemical reaction and experience its self-healing properties and stretchiness.	- Understand the principle of hydrogel formation and its properties. - Learn how to synthesize hydrogel.
v	Mushroom	- Prepare the substrate to grow mycelium-bound composites.	- Understand what is needed to grow fungi. - Learn how to prepare a mycelium-bound composite using natural elements.

**Table 3 biomimetics-08-00125-t003:** Summary of research questions and activities carried out during the workshop to answer thesis research questions.

Item	Research Question	Workshop Activity
1	What was the participants’ predisposition to learning about new forms of design that would lead to a more sustainable world, which is one end goal of biomimicry and bioinspiration?	Preworkshop survey Question 1: Can you give some examples of biomimicry around us and your thoughts about how this can be useful?
2	How do the participants understand and conceptualize the biomimicry and bioinspiration process in the top-down scenario?	Part 1 of the workshop Question 2a: What is the problem addressed? Question 2b: What natural system was chosen? Question 2c: How did they apply the natural solution to the engineering system?
3	How do the participants understand, conceptualize, and apply biomimicry and bioinspiration in the bottom-up scenario?	Part 2 of the workshop Question 3a: What initial thought came to mind when seeing the object? Question 3b: What did you do in the lab, and what key concept did you learn? Question 3c: For what application could the mechanism be learned to be used?
4	What was the learning experience and change of vision, if any, of the students?	Postworkshop survey

**Table 4 biomimetics-08-00125-t004:** Answers to Question 1 are classified into “nature-oriented” and “technology-oriented” groups. The bioinspiration trait column relates to the kind of biomimicry involved, and the field of application relates to the area in which the proposition could be applied. The tick marks indicate that the answer did not provide the bioinspiration trait or application area.

	Bioinspiration Trait	Field of Application
Nature-Oriented Applications of Biomimicry and Bioinspiration (Bottom-Up Process)		
The streamline[d] shape of ships [is] similar to fish’s (HS_1) Sticky tape mimicking the toes of geckos (HS_7) How ants are able to lift objects several times its [their] weight (HS_6) The lift on aircraft [is] similar to birds in flight (UG_5) The airfoil on airplanes is modeled after a bird’s wing (UG_10) The ways plants react to stimuli (UG_6) How rain is collected similar[ly] to plants (UG_14) Umbrellas from palm trees (UG_12) The lotus-shaped buildings in Singapore (UG_6) Shade from umbrella like the shape of trees (UG_13) The way we work together on anything like the life of ants (UG_9)	shape mechanism performance mechanism shape mechanism mechanism shape shape shape mechanism	underwater - - transportation transportation robotics - - construction - -
**Technology-Oriented Applications of Biomimicry and Bioinspiration (Top-Down Process)**		
High-speed rail (HS_3) Helicopter (HS_9) Night sight goggles (HS_10) Bats and radar (HS_2) Beehives design to maximize storage space (HS_4) Application in [the] construction sector (HS_6) Beehive structured buildings (HS_4) Velcro (HS_5) Military uses (HS_6) Swimsuit (HS_9) Needles (HS_11) Robots based on insects (UG_3) Subway system efficient routes (UG_4) UAVs inspired by birds and insects (UG_1) Green buildings (UG_7) Construction (UG_2) Hydrophobic materials (UG_1) Underwater equipment (UG_3) Artificial intelligence (UG_2) Detecting sunlight at different positions (UG_5) Low energy use structures (UG_11)	performance - new capabilities - sustainability - - - - - - - performance - sustainability - performance - - new capabilities sustainability	transportation transportation vision vision/communication energy construction construction textiles military underwater medicine robotics transportation transportation construction construction - underwater robotics - energy

**Table 5 biomimetics-08-00125-t005:** Answers to Questions 2a, 2b, and 2c. Examples of answers provided are given, and student IDs are indicated in parentheses.

Question 2a: What Is the Problem Addressed?		
	Application	Performance
Scenario A	“…loud noise that the bullet train makes when exiting a tunnel.” (HS_4) “The trains were creating very loud sonic booms when exiting tunnels.” (UG_5)	“The original train design created huge noise disturbances and was not very energy efficient.” (HS_8) “…inefficiency in [the] aerodynamics of a bullet train.” (UG_11)
Scenario B	“Desalination” (HS_1) “Water treatment and transport” (UG_8)	“Inefficient water filtering system” (UG_1)
**Question 2b: What natural system was chosen?**		
	**General**	**Specific**
Scenario A	“Birds” (HS_2) “Certain attributes of different birds were taken into consideration and copied” (UG_5)	“Owls feathers, penguins’ belly, kingfishers’ beak” (HS_6) “The beak of a kingfisher, the belly of a penguin, and the feathers of an owl were chosen.” (UG_12)
Scenario B	“Plants” (HS_7) “Trees” (UG_2)	“Osmosis in plants” (HS_1) “Aquaporin proteins seen in plants” (UG_8)
**Question 2c: How did they apply the natural solution to the engineering system**		
	**Product modification**	**Mimicking nature**
Scenario A	“Modified the shape of the head of [the] train to reduce noise pollution” (HS_3) “They were able to make a train that didn’t make the sonic booms when exiting a tunnel while keeping it extremely efficient” (UG_5)	“Shaped the nose of the train like the beak [of] the kingfisher” (HS_4) “The kingfisher beak shape applied to the locomotive increased speed and efficiency of the train while solving the sonic boom problem” (UG_13)
Scenario B	“Enhanced membrane in desalination system” (HS_4) “They created an artificial matrix with aquaporin channels in it” (UG_6)	“Designing a membrane similar to plant for better transportation of water” (HS_9) “Making the filters mimic the plant membranes to improve efficiency and sustainability” (UG_1)

**Table 6 biomimetics-08-00125-t006:** Answers to Questions 3a, 3b, and 3c. Examples of answers provided are given, and the students’ group IDs are indicated in parentheses.

Question 3a: What Initial Thought Came to Your Mind When Seeing the Biological Sample?	
Description	Comparison with Existing
“White stuff at the base” (HS_v) “Spiral shaped” (HS_iii) “Soft and sticky” (HS_iv) “Helix shaped” (HS_i) “Curved” (HS_i) “Gummy” (UG_iv) “Brightly colored” (UG_iv) “Fragile” (UG_v)	“Gummy bear” (HS_iv) “Umbrella” (HS_ii) “Rainwater catchment.” (HS_ii) “Food” (HS_v) “Sap-like” (UG_iv)
**Question 3b: What did you do in the lab, and what key concept did you learn?**	
**Technical/Scientific Tools and Methods**	**Technical/Scientific Concepts**
“Portable microscope” (HS_ii) “Aseptic techniques” (HS_v) “Vickers micro indenter” (HS_iii) “Crosslinking” (HS_iv) “3D printing.” (HS_i) “Contact angle” (UG_ii) “Applied force” (UG_iii) “Sanitized.” (UG_v)	“Anisotropy property” (HS_iii) “We learned that the elasticity of the gels varies with the temperature.” (HS_iv) “Hydrophobic properties” (UG_ii) “More jagged crack that dissipates energy more efficiently than a straight crack” (UG_iii)
**Question 3c: For what application could the mechanism learn to be used?**	
**General Application**	**Specific Technical Solution**
“Self-cleaning surface” (HS_ii) “Packaging” (HS_v)“To upcycle non-degradable items” (HS_v) “Cosmetic surgery, temporary adhesive (replace glue tack, glue sticks).” (HS_iv) “Waterproofing clothing or other products” (UG_ii) “Construction, airplanes” (UG_iii) “Seal a hole or break in a pipe or bottle.” (UG_iv) “Reuse waste material.” (UG_v)	“Control of properties dependent on direction” (HS_iii) “Printing chemically similar materials to follow similar property styles.” (HS_i) “Replicate the rough microstructure in the clothing products.” (UG_ii) “A synthetic nacre-like structure” (UG_iii) “Heating up and molding the material around the place in need of sealing” (UG_iv)

## Data Availability

The data presented in this study are available on request from the corresponding author. The data are not publicly available due to confidentiality.

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
