# Peer review of "Conceptualization of Biomimicry in Engineering Context among Undergraduate and High School Students: An International Interdisciplinary Exploration"

_biomimetics, 2023, doi:10.3390/biomimetics8010125_

Round 1

Reviewer 1 Report

Please see a few suggestions and minor edits in the document attached.

Reviewer 2 Report

In this study, the authors have explored the difference in top-down and bottom-up bioinspired design approaches in terms of design conceptualization. The authors conclude that bottom-up approach is more easily understood by engineering students as compared to a top-down approach. A bottom-up approach enables students to cognitively focus on specific details of a biological entity.

The presented findings are interesting and novel. However, the manuscript needs some revisions before it is accepted for publication. In particular, this study requires more evidence for the claims, more writing focus on the novel contributions and some more content to improve reproducibility of this research.

The following is my feedback on the manuscript:

Line 34: While I understand that the terms ‘bioinspiration’ and ‘biomimicry’ are used interchangeably, I would suggest the authors to use the terminology in ISO 18458 for clarifying the reader on what the authors use of there terms are.

There are some statements in the manuscript that are strongly generalized and would require more evidence. For example, (Line 43) “Contrary to more traditional scientific research, biomimicry and bioinspiration tend to focus on environmental and social impacts while still providing performance and economic outputs.” I would suggest giving some examples to support this claim.

The authors have done a good job quantifying the trends in the literature to justify this work’s importance. It is unclear to me how the layout (questions/lab) for this study were setup keeping in mind the limitations of the existing studies. In other words, what particular features of this studies were designed such that the limitations from the listed studies are not repeated?

Line 192, 198: I did not understand the context of “hot earth to cold space” and “no nature in space”. From what I get, the lecture tried to apply bioinspiration to space applications, but the exact question or the example application needs to be clarified.

Line 275: As independent coders coded (categorized) the data responses, it would be good to know the inter-rater agreement between them to establish the reliability of the categorization.

Line 294: confidence in Western undergraduates is more than high-school students. Could this largely be due to difference in their technical skillset?

I had a hard time finding and going back and forth for the question numbers that have been referred to in various parts of the manuscript. I would suggest creating a table that contains all the questions.

I believe the manuscript word count should be reduced by reducing the focus on less significant content. For example, Line 265: about how participants responded to questions is not significant unless it was specifically designed to reduce bias or misunderstanding.

In addition, I believe manuscript should focus on the most novel findings from this study. Focus on peripheral findings should be reduced to improve the impact of this study. Similarly, the authors can focus less on the validation of other studies’ findings and more on the cognitive behavior difference in top-down and bottom-up approaches.

Line 493: I am not sure that the ‘need for interdisciplinarity’ described here is a significant contribution of this manuscript. Engineering-biologist collaboration was not performed in this study and the limitations listed are the findings of the prior studies.

One important point that may affect this study’s results is the chronology of teaching top-down and bottom-up approaches. Could the students have grasped bottom-up approach better because the prior teaching of the top-down approach served as an ice-breaker/warm-up. Likewise, the prior exposure of the students to bioinspired design may also impact the result. I would suggest reflecting on these in your manuscript.

The conclusions and recommendations need to more to-the-point. Topics that need to be described here are what the high-level impact of this study is, the observed biases (constructive/deprecative) and how future studies can use the findings of this work. For example,a top-down approach is desirable for engineers, but this study shows that it is difficult to grasp by the students. So, what are the recommendations from this study to enable a top-down approach?
